# Health and well-being of male international migrants and non-migrants in Bangladesh: A cross-sectional follow-up study

**Randall Kuhn**[1]*, **Tania Barham**[2], **Abdur Razzaque**[3], **Patrick Turner**[4]

**1** Department of Community Health Sciences, UCLA Fielding School of Public Health, Los Angeles, California, United States of America, **2** Department of Economics, University of Colorado Boulder, Boulder, Colorado, United States of America, **3** Health and Population Surveillance Division, icddr,b, Dhaka, Bangladesh, **4** Department of Economics, University of Notre Dame, Notre Dame, Indiana, United States of America

* kuhn@ucla.edu

**Data Availability Statement:** Data will be made available for public release through the Data Sharing for Demographic Research portal of the

## Abstract

### Background

Temporary labor migration is an increasingly important mode of migration that generates substantial remittance flows, but raises important concerns for migrant well-being. The migration and health literature has seen a growing call for longitudinal, binational surveys that compare migrants to relevant non-migrant counterfactual groups in the sending country, in order to answer the basic question "Is migration good for health?" This study compares the health of male international migrants, internal migrants, and non-migrants using a unique representative panel survey of the Matlab subdistrict of Bangladesh.

### Methods and findings

A cohort of 5,072 respondents born 1958–1992 were interviewed in 1996–1997, and reinterviewed in 2012–2014. Extensive migrant follow-up yielded a 92% reinterview rate. We explored health and income outcomes for respondents who at the time of the follow-up interview were current international migrants ($n = 790$), returned international migrants ($n = 209$), internal migrants ($n = 1,260$), and non-migrants ($n = 2,037$). Compared to non-migrants, current international migrants were younger (mean 32.9 years versus 35.8 years), had more schooling (7.6 years versus 5.8 years), and were more likely to have an international migrant father (9.7% versus 4.0%) or brother (49.1% versus 30.3%). We estimated multivariate ordinary least squares and logistic regression models controlling for a wide range of control variables measured as far back as 1982. Results show that current international migrants had substantially better health status on factors that likely relate to self-selection such as grip strength and self-rated health. Current international migrants had no excess risk of injury in the past 12 months compared to non-migrants (adjusted mean risk = 6.0% versus 9.3%, $p = 0.084$). Compared to non-migrants, current international migrants had roughly twice the risk of overweight/obesity (adjusted mean risk = 51.7% versus 23.3%, $p < 0.001$), obesity (6.9% versus 3.4%, $p = 0.012$), and stage 1 or higher hypertension (13.0% versus 7.0%, $p =$

Interuniversity Consortium for Political and Social Research, under the requirements of US National Institute of Child Health and Human Development data archiving grant R03 HD093987-01 "The Matlab Linked Database: A 40-year archive of health, population and development data in rural Bangladesh." The data can be found at: https://www.icpsr.umich.edu/icpsrweb/NACDA/studies/2705/version/1. MHSS2 data will available for public download at DSDR, enabling replication of most analyses. The full dataset including MHSS1 and 1982 census baseline controls will be available via restricted access request through ICPSR. This is necessary because earlier versions of these older data are subject to deductive disclosure risks. Therefore the MHSS2 data cannot be linked to them without IRB protection. The only restriction on access to the crosswalk linking MHSS2 to baseline data will be provision of proof of IRB approval from the home institution. Requests will be handled impartially by DSDR and not be the study team, eliminating unnecessary barriers to access. All code used in completing this paper will be released along with this archive.

**Funding:** This work was funded by Research & Empirical Analysis of Labor Migration (REALM) Program Grant 20164975, under a grant from the New York University Abu Dhabi with Peter S. Bearman from Columbia University acting as Program Director; National Institute on Aging Grant 1R01AG033713-01A1 (Jane Menken, PI); and International Initiative for Impact Evaluation (3ie), Open Window Round 3 Grant OW3.1060, "Thirty - Five Years Later: Evaluating Effects of a Quasi-Random Child Health and Family Planning Program in Bangladesh". The corresponding author benefited from facilities and resources provided by the California Center for Population Research at UCLA (CCPR), which receives core support (P2C-HD041022) from the Eunice Kennedy Shriver National Institute of Child Health and Human Development (NICHD). No funding bodies had any role in study design, data collection and analysis, decision to publish, or preparation of the manuscript.

**Competing interests:** The authors have declared that no competing interests exist.

**Abbreviations:** CES-D, Center for Epidemiologic Studies Depression Scale; GCC, Gulf Cooperation Council; kgf, kilograms-force; MHDSS, Matlab Health and Demographic Surveillance System; MHSS, Matlab Health and Socioeconomic Survey.

0.014). Compared to internal migrants, current international migrants had significantly higher levels of overweight/obesity (adjusted mean risk = 51.7% versus 37.7%, $p < 0.001$). Current international migrants showed above average levels of depressive symptoms on a 12-item standardized short-form Center for Epidemiologic Studies Depression Scale (+0.220 SD, 95% CI 0.098–0.342), significantly higher than internal migrants (−0.028 SD, 95% CI −0.111, 0.055; $p < 0.001$). Depressive symptoms differed significantly from those reported by non-migrants when restricting to items on negative emotions (international migrant score = 0.254 SD, non-migrant score = 0.056 SD, $p = 0.004$). Key limitations include the descriptive nature of the analysis, the use of both in-person and phone survey data for international migrants, the long recall period for occupational and mental health risk measures, and the coverage of a single out-migration area of origin.

## Conclusions

In this study, we observed that international migrants had comparable or lower injury and mortality risks compared to respondents remaining in Bangladesh, due in part to the high risks present in Bangladesh. International migrants also showed higher levels of self-rated health and physical strength, reflective of positive self-selection into migration. They had substantially higher risks of overweight/obesity, hypertension, and depression. Negative health impacts may reflect the effects of both harsh migration conditions and assimilation into host population conditions. Our results suggest the need for bilateral cooperation to improve the health of guest workers.

## Author summary

### Why was this study done?

- Temporary international guest workers comprise perhaps one-third of the world's migrant workers, or about 50–60 million in 2018. Concerns have been raised about the health and safety of these workers, particularly those working in the nations of the Gulf Cooperation Council (GCC).

- Yet little rigorous evidence has quantified these risks, and guest workers typically migrate from highly disadvantaged sending areas where health risks might be equally high or higher.

- In recent years, studies of migrant well-being have recognized the need to compare migrants to relevant non-migrant counterfactual groups, in order to answer the basic question "Is migration good for your health?"

### What did the researchers do and find?

- In 1996–1997, we interviewed 5,072 male respondents born 1958–1992. In 2012–2014, we successfully located 92% of surviving respondents ($n = 4,296$), and classified respondents as current international migrants ($n = 790$), returned international migrants ($n = 209$), internal migrants (1,260), and non-migrants (2,037). More than 70% of international migrants worked in GCC countries.

- We estimated the metabolic health, mental health, injury risk, and income of the 4 migration status groups, controlling for a wide range of factors relating to current socio-demographic characteristics, family background, and familial migration history. We report results in terms of "adjusted" means to allow easy comparison of the risks facing international migrants to those of other groups.

- International migrants showed comparable or lower levels of injury and mortality risk than other groups. Migrants also fared better on physical strength and general self-rated health.

- International migrants had much higher risk of being overweight/obese (52%) than non-migrants (23%) and internal migrants (38%), they had higher risk of obesity (6.9%) than non-migrants (3.4%), and they had higher risk of stage 1 or higher hypertension (13.0%) than non-migrants (7.0%). International migrants also reported a significantly higher level of depressive symptoms than non-migrants and internal migrants.

### What do the findings mean?

- Many concerns have been raised about the health and safety of guest workers, particularly those living in the Gulf Cooperation Council countries. Guest workers do face excess health risks compared to those who do not migrate, but these may not be the risks that draw widespread media attention.

- While injury and mortality risks were lower and migrants were far stronger physically, they faced higher burdens of obesity, hypertension and mental illness that may reflect both the harsh conditions of migration and some adoption of local behaviors of the host country.

## Introduction

Temporary labor migration is an increasingly important mode of migration that generates substantial remittance flows, but has important consequences for migrant well-being [1,2]. Up to one-third of the world's migrant workers are thought to work on temporary visas, meaning about 50–60 million workers in 2018 [1,3]. Advocates and the popular press have raised concerns about mortality, injury, mental health, and life course health risks facing these workers, especially those living and working in the nations of the Gulf Cooperation Council (GCC) [4,5]. However, guest workers typically migrate from highly disadvantaged sending areas where health risks may be equally high or higher [6]. To date, no large-scale survey to our knowledge has systematically documented health outcomes among a large, representative cohort of guest workers, or compared them to outcomes among non-migrant counterfactuals. This study compares the health of international migrants, internal migrants, and non-migrants using a unique representative panel survey of the Matlab subdistrict of Bangladesh, an area characterized by unusually high rates of migration to labor-importing countries in the Persian Gulf and Southeast Asia.

In recent years, studies of migrant well-being have recognized the importance of comparing migrants to relevant non-migrant counterfactual groups, in order to answer the basic question "Is migration good for your health?" along with a host of questions relating to specific health

conditions, risk factors, and structural mechanisms [7–10]. Due to data limitations, most studies comparing the health of emigrants to that of non-migrants take the innovative but limited approach of pooling or matching separate datasets from sending and receiving countries [8,11,12]. Given the methodological challenges facing these studies and the continued struggle to account for self-selection across context, these studies have yet to give rise to a systematic or generalizable set of conclusions regarding the health consequences of migration for migrants [13,14]. There has thus been a growing call for binational panel surveys that can measure the conditions, pre- and post-migration, affecting a set of self-selected international migrants who are drawn from a large representative sample of an origin population, in which non-migrants and internal migrants serve as comparison groups [15]. Yet existing binational migrant surveys are limited by small [16,17] or purposive migrant samples [14,18].

Evidence on the health of migrants to the Persian Gulf region is limited due to the difficulty of collecting data in the receiving countries. Studies of occupational injuries noted rates of injury and fatality above those experienced by host populations, but did not compare rates to those in the countries of origin or in high-risk occupations in other countries [19–21]. A recent systematic review of all studies related to migrant mental health in the Persian Gulf identified fewer than a dozen articles published between 2002 and 2010 [22]. These limited studies offer quantitative evidence of a high rate of suicide [23–26] and psychiatric symptoms [27,28]. A small number of studies have used depression scales to quantify burdens and covariates of mental illness among migrants [29,30]. Even fewer studies have addressed general health or metabolic health, though these tend to identify high burdens among migrants [31,32]. None of these studies was longitudinal or included a non-migrant reference group. A larger body of studies documents the existence of social and occupational conditions that may give rise to health risk, including legal vulnerability [33–36], occupational abuse [37–39], and indebtedness [40].

Using 2 rounds of survey data from the Matlab Health and Socioeconomic Survey (MHSS), we compare the health and livelihoods of temporary international migrants to those of non-migrants and internal migrants interviewed in 2012–2014 who were drawn from the same representative sample initially interviewed in 1996–1997. We look at effects on injury, general health, hypertension, mental health, and risk factors such as smoking and obesity.

## Methods

The study used 2 rounds of panel survey data from the MHSS, with the first round, MHSS1, conducted in 1996–1997 and the second round, MHSS2, conducted in 2012–2014. The rural Matlab subdistrict is the site of the Matlab Health and Demographic Surveillance System (MHDSS), the longest-running vital registration system in the Global South, instituted by icddr,b (International Centre for Diarrhoeal Disease Research, Bangladesh). MHDSS provided prospective records from of all birth, death, marriage, and migration episodes in a 141-village study area. MHDSS offers numerous advantages for the collection of high-quality health survey data, including precise age data for individuals born since 1974, prospective observation of prior demographic events including migration and mortality, and long-term baseline data from censuses conducted in 1974 and 1982. In 1996–1997, MHSS1 was conducted by icddr,b, RAND, University of Pennsylvania, and partner institutions, with funding from the National Institute on Aging and the National Institute of Child Health and Human Development. MHSS1 collected detailed health, social, demographic, and economic data for 2,637 MHDSS households, a 7% random sample of the study area. MHSS1 included data on family migration history, household assets, kinship, remittances, schooling, and cognition at baseline.

Conducted from 2012 to 2014, MHSS2 followed all individuals from MHSS1. Protection of human participants during fieldwork and data analysis was ensured under icddr,b Ethical Review Committee Protocol #PR-10005. Fieldwork included an unusually long and intensive migrant follow-up effort [41]. A migrant's last known location was identified first through the electronic records in MHDSS, and then through intensive migrant tracking conducted in the last known household and with other kin. Migrant interviews were carried out in 4 phases: (1) as part of the regular fieldwork, (2) through a rapid response method aimed at maximizing familial connections to quickly complete interviews with migrants, (3) through interviews with returning international migrants and migrants in faraway domestic locations during the Muslim Eid festivals, and (4) through a phone survey that captured a subset of key survey questions [41].

The analysis was carried out with support from the Research & Empirical Analysis of Labor Migration (REALM) program administered by New York University Abu Dhabi. The analysis met all requirements of the Strengthening the Reporting of Observational Studies in Epidemiology (STROBE) guidelines, as shown in S1 STROBE Checklist. The analysis plan (included as S1 Text) identified the (1) statistical methodology, (2) coding scheme for the migration status treatment variable, and (3) control variables, but it did not name the specific dependent variables. Analysis followed the exact guidelines of the protocol, with the exception of restricting the sample to a narrower age range in order to focus on age groups with a non-trivial risk of current international migration. Analysis focused on 5,072 males born 1958 to 1992, who were age 20–56 years at the time of MHSS2. Fig 1 reports survival and interview follow-up status stratified by the respondent's last known location. Just over half were living in Matlab or the surrounding Chandpur/Comilla districts (including never-migrants and returned international migrants; 54%, 2,727 cases), 28% were outside these districts (internal migrants; 1,400 cases), and 19% were outside the country (international migrants; 945 cases). The proportion dying over the ≥16-year follow-up period was 2.6%. Death was more likely among respondents who were non-migrants at the time of survey (3.7%) compared to internal migrants (1.9%) or current international migrants (0.4%).

The follow-up rate was unusually high for a 16- to 18-year follow-up, with 92% of survivors interviewed and high rates of follow-up for non-migrants (94%), internal migrants (89%), and international migrants (90%). The resulting interview sample of 4,545 respondents was 54% non-migrant, 27% internal migrant, and 19% international migrant. Of the 4,545 eligible respondents, 240 (5% of total) were excluded due to missing variables in 1 of the 2 rounds, yielding a sample of 4,296 respondents. Missing data were spread across a wide range of variables, primarily income and baseline variables.

Statistical estimates of health and economic outcomes at MHSS2 were based on weighted ordinary least squares regression (for continuous variables) and weighted logistic regression (for binominal variables) controlling for migration status at MHSS2, time-invariant individual characteristics (year of birth, religion), current individual characteristics measured in MHSS2, life course individual characteristics in MHSS1, parental schooling, sibling composition, father's and brother's migration history, and baseline characteristics measured in 1982. We report our estimates of the association of migration status with health and economic variables in terms of the marginal predicted values (for continuous variables) or marginal predicted probabilities (for binomial variables). Along with predicted values/probabilities and confidence intervals, we report statistical tests of difference in which current international migrants are the reference category, thereby comparing international migrants to all 3 counterfactual comparison groups: non-migrants, internal migrants, and returned international migrants. Coefficient estimates for all control variables are reported in S1 Table for the model specifications shown in the main tables. We also estimated 2-stage propensity score models in which

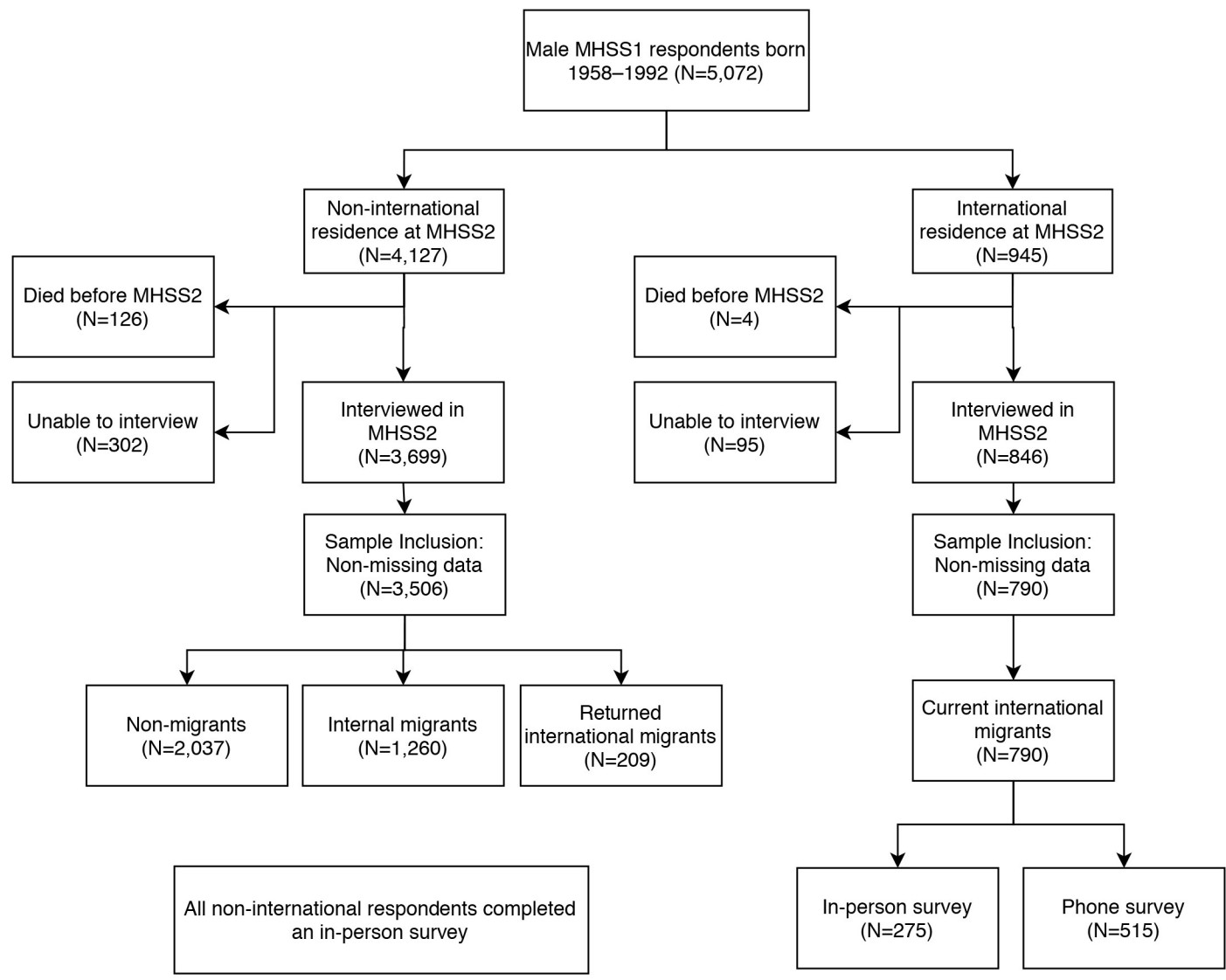

**Fig 1. MHSS1 to MHSS2 follow-up and sample inclusions.**

baseline variables were used to predict selection into international migration. Because this approach did not alter the results, we report only the more intuitive single-stage estimates.

## Dependent variables

We explored how migration impacts a broad range of health outcomes. For all respondents, including phone survey respondents, we tested models of self-rated health, injury and morbidity, overweight/obesity risks based on height/weight (observed for in-person interviews, self-reported for phone interviews), and a shortened 12-question version of the Center for Epidemiologic Studies Depression Scale (CES-D). Because there is no validated diagnostic cutoff for depression using this abridged scale, we simply estimated models for standardized depressive symptom scores. Because items relating to positive emotions (e.g., happiness, hopefulness) had shown lower item–overall score correlations, particularly for migrants, we also tested models

separately for the 4 items relating to positive emotions and the 8 items relating to negative emotions (e.g., sadness, loneliness).

We explored differences in BMI, and followed the Asian reference standards [42] to further model overweight/obese (BMI $\geq$ 23 kg/m$^2$), obese (BMI $\geq$ 27.5 kg/m$^2$), and underweight (BMI $<$ 18.0 kg/m$^2$). Height and weight were directly observed for in-person respondents and self-reported in the international migrant phone survey. We observed minimal differences in BMI for international migrants between in-person and phone surveys (0.47 kg/m$^2$, or 0.14 SD). We reestimated BMI models excluding or adjusting phone survey respondents, with no qualitative change in the results. For respondents interviewed in person (including one-third of international migrants), we also conducted physical examinations and measured whether a respondent had stage 1 or higher hypertension (defined as systolic blood pressure $>$ 130 mm Hg or diastolic blood pressure $>$ 80 mm Hg per the 2017 ACC/AHA guidelines [43]) and tested mean grip strength across 2 trials (measured in kilograms-force [kgf]).

We measured annual income and hours worked directly using a recall matrix that asked respondents to report work across a wide range of job types (e.g., salaried employment, daily work, agricultural work).

### Independent variable

The migration status measure compared current international migrants, returned international migrants, current internal migrants, and non-migrants. Migration status was identified based on current place of residence. Among respondents in Bangladesh, returned international migrants were identified based on a self-report of whether they had lived abroad in the past 5 years. Internal migrants were identified based on area of residence outside Chandpur/Comilla districts, with the majority moving for work to major cities such as Dhaka and Chittagong, and others moving to nearby towns. All remaining respondents were classified as non-migrants. For simplicity of presentation, we assigned respondents to mutually exclusive migration status categories, with returned international migrant status superseding current residential status. The analytic sample includes 4,296 respondents, with 790 living internationally at the time of follow-up (referred to as "current international migrants"), 2,037 non-migrants, 1,260 internal migrants, and 209 returned international migrants. Of the 790 current international migrants, 515 were interviewed by phone and 275 in person.

The statistical estimates reported in the main tables come from regressions that also controlled for age, education, religion, height, father and mother's schooling (measured directly or reported by respondent), sibling composition, whether the respondent's father and any brother ever lived abroad (based on MHDSS tracking), and 1996–1997 household assets (taken from MHSS1 as the sum value of assets across all productive and non-productive types). Religion was coded as 1 if the respondent was Hindu (9.5% of respondents) and 0 if the respondent was Muslim. Years of schooling were collected via survey, with individuals who never attended school or attended only Koranic recitation (Maktab) coded with 0 completed years. S1 Table also reports additional models that introduced controls for 1982 baseline variables (from the 1982 MHDSS census). Unreported models also controlled for (1) MHSS2 month of interview effects, (2) birth year fixed effects, and (3) village fixed effects. Each of these specifications yielded models that were comparable to the reported models.

## Results

### Bivariate analysis

Table 1 provides some background on countries of destination and living/working conditions for migrants in those destinations. Respondents primarily lived in Saudi Arabia (26%), United

**Table 1. Selected migrant living/working characteristics for international migrants, by destination country.**

| Characteristic | Saudi Arabia (*n* = 202) | United Arab Emirates (*n* = 204) | Other GCC country (*n* = 153) | Singapore or Malaysia (*n* = 171) | Other (*n* = 60) | Total (*n* = 790) |
|---|---|---|---|---|---|---|
| Has legal documentation | 92% | 93% | 92% | 94% | 92% | 93% |
| Used manpower agent | 97% | 97% | 98% | 98% | 93% | 97% |
| Keeps passport | 10% | 10% | 18% | 19% | 42% | 17% |
| Reads/writes local language | 7% | 2% | 1% | 13% | 17% | 6% |
| Lives in company housing | 47% | 55% | 38% | 56% | 49% | 49% |

Source: MHSS2 (2012–2014).

GCC, Gulf Cooperation Council.

Arab Emirates (26%), other GCC countries (19%), and Singapore or Malaysia (22%), with another 4% scattered in other guest worker destinations (e.g., Libya, Lebanon, Maldives) and the remaining 3% in Europe or North America. Table 1 reports the unique living and working conditions affecting migrants. The vast majority of migrants across all destinations (93%) had legal documentation, typically in the form of a 3-year temporary work permit. An even higher proportion had used a manpower agency to arrange their migration (97%). Few migrants were in possession of their own passports (17%), with the lowest share in Saudi Arabia (10%) and United Arab Emirates (10%) and the highest share in other destination countries (42%). Only a small percentage could read and write the local language, ranging from just 1% in other GCC to countries to 17% in other destination countries. About half of respondents lived in company housing, ranging from 38% in other GCC countries to 56% in Singapore/Malaysia.

Fig 2 summarizes variation in health outcomes by migration status from bivariate analysis. Current international migrants were significantly less likely than all groups to report fair/poor self-rated health (4% compared to 19% for non-migrants and 12% for internal migrants and returned international migrants). They also reported significantly lower levels of injury in the past year (5% versus 6%–9% for other groups), current smoking (30% versus 36%–39% for other groups), and poor grip strength (13% versus 23%–38% for other groups). Current international migrants showed substantially higher burdens of overweight/obese status (60%) in comparison to returned international migrants (46%), internal migrants (36%), and, especially, non-migrants (22%). Similar patterns emerge for obesity, though at much lower levels (10% for current international migrants versus 3%–7% for other groups). Current international migrants also had higher levels of hypertension (14% versus 8%–11% for other groups).

Table 2 compares the baseline characteristics of migration status groups. On most characteristics, current international and internal migrants were comparable, making internal migrants a relevant reference group. Current international migrants were younger than non-migrants (mean 33 versus 36 years), though internal migrants were even younger (mean 31 years). Current international migrants also had higher levels of schooling than non-migrants (mean 7.6 versus 5.8 years), but they were similar to internal migrants (mean 7.9 years). Current and returned international migrants were about 2 cm taller on average than non-migrants and 1 cm taller than internal migrants. While sibling composition was broadly similar across all categories, international migrants had preexisting familial socioeconomic advantages. International migrants had significantly higher levels of mother's and father's schooling relative to non-migrants, though significantly lower than those of internal migrants. International migrants were much more likely to have a father who had lived abroad or a brother who had lived abroad, though even 30% of non-migrants had an international migrant brother. Finally, both current and returned international migrants came from households whose 1996–1997 household assets were about 30% higher than those of both non-migrant and internal migrant households.

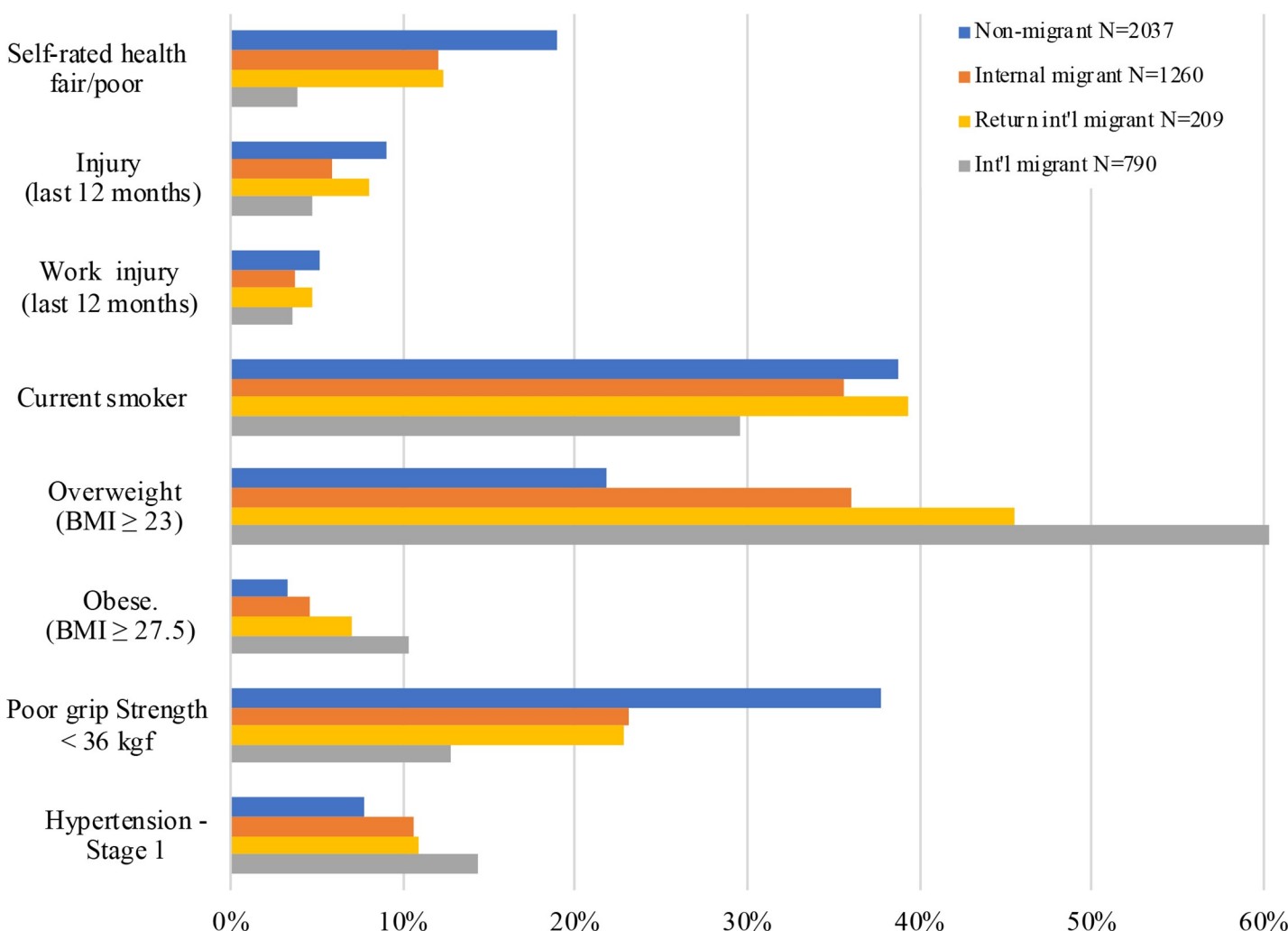

**Fig 2. Prevalence of key health outcomes, by migration status—Unadjusted.** For poor grip strength (<36 kilograms-force [kgf]) and stage 1 or higher hypertension, current international migrants include only those who were interviewed in person (*n* = 275).

## Multivariate analysis: Employment

We estimated differences in earnings, hours worked, and hourly wages, shown in Table 3. In terms of annual earnings, current international migrants earned about 4 times more than non-migrants (mean $5,032 versus $1,235, *p* < 0.001) and nearly 3 times more than internal migrants (mean $5,032 versus $1,813, *p* < 0.001). Work effort was high for all groups, but current international migrants worked substantially more hours per week than non-migrants (mean 66.8 versus 48.6), though only about 10% more than internal migrants (mean 60.8). Returned international migrants earned comparable (though slightly higher) wages than non-migrants on similar hours worked.

## Multivariate analysis: Physical health

Table 4 evaluates the association between migration status and health measures after controlling for baseline variables. After controlling for other factors, current international migrants remained significantly less likely to report fair/poor self-rated health status (6.7%) than non-

**Table 2. Baseline characteristics of respondents, by migration status.**

| Characteristic | Non-migrant $n = 2,037$ | Internal migrant $n = 1,260$ | Returned international migrant $n = 209$ | Current international migrant $n = 790$ | Total $n = 4,296$ |
|---|---|---|---|---|---|
| Age (years) | 35.8 (10.4) | 30.8 (8.2) | 34.4 (7.2) | 32.9 (6.7) | 33.8 (9.3) |
| Years of schooling | 5.8 (4.6) | 7.9 (4.6) | 7.7 (3.8) | 7.6 (3.3) | 6.8 (4.5) |
| Religion Hindu | 13.3% | 7.6% | 4.3% | 4.0% | 9.5% |
| Height (cm) | 162.9 (6.4) | 163.8 (6.0) | 164.5 (6.0) | 164.7 (5.0) | 163.6 (6.1) |
| Younger brothers | 1.0 (1.1) | 0.9 (1.0) | 1.0 (1.0) | 1.1 (1.1) | 1.0 (1.1) |
| Older brothers | 1.0 (1.2) | 1.1 (1.3) | 1.2 (1.4) | 1.2 (1.3) | 1.1 (1.3) |
| Younger sisters | 1.0 (1.1) | 0.9 (1.0) | 0.9 (1.0) | 0.9 (1.0) | 1.0 (1.1) |
| Older sisters | 0.9 (1.1) | 0.9 (1.1) | 1.1 (1.2) | 1.0 (1.2) | 0.9 (1.1) |
| Father's years of schooling | 3.0 (3.7) | 3.8 (4.2) | 3.6 (3.9) | 3.3 (3.7) | 3.3 (3.9) |
| Mother's years schooling | 1.1 (2.2) | 1.9 (2.8) | 1.5 (2.2) | 1.5 (2.4) | 1.5 (2.5) |
| Father lived abroad | 4.0% | 5.9% | 6.7% | 9.7% | 5.8% |
| Brother lived abroad | 30.3% | 21.2% | 51.2% | 49.1% | 32.1% |
| Household assets, 1996–1997 (US dollars) | 4,858 (7,962) | 4,813 (7,432) | 6,228 (7,110) | 6,397 (8,579) | 5,150 (7,878) |

Data are given as mean (SD) or percent. Source: MHSS1 (1996–1997), MHSS2 (2012–2014), Matlab Health and Demographic Surveillance System (1982–2014).

migrants (16.6%, $p < 0.001$), internal migrants (12.8%, $p = 0.012$), and returned international migrants (13.8%, $p = 0.30$). Current international migrants were less likely than non-migrants, though not significantly so, to have experienced an injury in the past 12 months (6.0% versus 9.3%, $p = 0.084$). We found no significant differences between current international migrants and other groups in the likelihood of having a work-related injury, and, after controlling for covariates, migrant advantages in smoking were no longer significant.

The introduction of controls also diminished the magnitude and significance of the association of migration with overweight/obese (BMI $\geq$ 23.0 kg/m$^2$) and obese (BMI $\geq$ 27.5 kg/m$^2$) status, yet effects remained significant. After adjustments, mean BMI for current international migrants was 23.3 kg/m$^2$ (95% CI 23.0–23.6), with highly significant differences from non-migrants (mean 20.9 kg/m$^2$, 95% CI 20.7–21.0, difference = 2.4 kg/m$^2$, $p < 0.001$) and internal migrants (mean 22.0 kg/m$^2$, 95% CI 21.7–22.2, difference = 1.3 kg/m$^2$, $p < 0.001$). BMI for returned international migrants was between that of current international migrants and internal migrants (mean 22.7 kg/m$^2$, 95% CI 22.0–23.3), with the difference from current international migrants significant only at the 10% level. International migrants remained more than

**Table 3. Estimates of earnings, hours, and hourly wages—covariate-adjusted ordinary least squares regression models.**

| Measure | Non-migrant | Internal migrant | Returned international migrant | Current international migrant | $n$ |
|---|---|---|---|---|---|
| Annual income | $1,235*** | $1,813*** | $1,500*** | $5,032 | 4,296 |
| | ($1,036–$1,407) | ($1,607–$2,019) | ($1,081–$1,919) | ($4,654–$5,411) | |
| Hours worked per week | 48.6*** | 60.8*** | 48.5*** | 66.8 | 4,296 |
| | (47.0–50.2) | (59.0–62.7) | (43.4–53.6) | (64.9–68.8) | |
| Hourly wage | $0.51*** | $0.64*** | $0.64*** | $1.49 | 4,047 |
| | ($0.45–0.57) | ($0.56–0.71) | ($0.48–0.79) | ($1.38–$1.60) | |

Data are mean (95% CI). Marginal predictions from regressions controlling for age, education, religion, height, parental schooling, sibling composition, father's international migration, brother's international migration, and 1996–1997 household assets. Statistical test of difference from current international migrant
***$p < 0.001$.

**Table 4. Estimates of health outcomes by migration status from covariate-adjusted logistic regression models.**

| Measure | Non-migrant | Internal migrant | Returned international migrant | Current international migrant | n |
|---|---|---|---|---|---|
| **Self-reported health measures** | | | | | |
| Fair/poor self-rated health | 16.6%** | 12.8%* | 13.8%* | 6.7% | 4,296 |
| | (14.4%–18.7%) | (10.1%–15.6%) | (7.6%–20.0%) | (3.6%–9.9%) | |
| Injury | 9.3%+ | 5.1% | 8.7% | 6.0% | 4,296 |
| | (7.6%–11.0%) | (3.4%–6.8%) | (3.5%–13.8%) | (3.3%–8.8%) | |
| Work-related injury | 5.2% | 3.5% | 6.2% | 5.3% | 4,296 |
| | (4.0%–6.5%) | (2.0%–5.0%) | (1.4%–10.9%) | (2.5%–8.0%) | |
| Current smoker | 37.6% | 37.7% | 37.9% | 32.7% | 4,296 |
| | (34.6%–40.5%) | (33.7%–41.7%) | (29.2%–46.5%) | (27.7%–37.7%) | |
| Body mass index | 20.9*** | 22.0*** | 22.7+ | 23.3 | 4,296 |
| | (20.7–21.0) | (21.7–22.2) | (22.0–23.3) | (23.0–23.6) | |
| Overweight or obese | 23.3%*** | 37.7%** | 47.0% | 51.7% | 4,296 |
| | (20.7%–26.0%) | (33.8%–41.6%) | (37.5%–56.5%) | (46.5%–56.9%) | |
| Obese | 3.4%* | 5.1% | 5.0% | 6.9% | 4,296 |
| | (2.2%–4.7%) | (3.3%–7.0%) | (2.0%–8.1%) | (4.5%–9.2%) | |
| Underweight | 17.5%*** | 11.2%*** | 7.8%** | 2.2% | 4,296 |
| | (15.2%–19.8%) | (8.9%–13.4%) | (2.7%–12.8%) | (0.7%–3.7%) | |
| **Objective health measures (in-person interviews only)** | | | | | |
| Mean grip strength (kgf) | 38.6*** | 39.6*** | 40.0* | 41.8 | 3,754 |
| | (38.3–39.0) | (39.1–40.2) | (38.8–41.3) | (40.7–42.8) | |
| Hypertension—stage 1 or higher | 7.0%* | 12.6% | 8.4% | 13.0% | 3,760 |
| | (5.6%–8.4%) | (9.8%–15.5%) | (3.8%–12.9%) | (7.4%–18.6%) | |

Data are mean (95% CI). Marginal predictions from regressions controlling for age, education, religion, height, parental schooling, sibling composition, father's international migration, brother's international migration, and 1996–1997 household assets. Statistical test of difference from current international migrant

***$p < 0.001$

**$p < 0.01$

*$p < 0.05$

+$p < 0.10$.

kgf, kilograms-force.

twice as likely as non-migrants to be overweight/obese (51.7% versus 23.3%, $p < 0.001$) or obese (6.9% versus 3.4%, $p = 0.012$) than non-migrants. International migrants were also significantly more likely to be overweight/obese than internal migrants (51.7% versus 37.7%), though differences in obesity were not significant. Underweight status also varied considerably across the groups, with very few current international migrants observed as underweight (mean 2.2%, 95% CI 0.7%–3.7%), a rate significantly lower than those of all other groups.

Objective health measures were not included in the phone survey, and so analysis of these variables includes only the 275 current international migrants interviewed during the Eid festivals. In spite of this small sample size, we observed significant differences. Current international migrants achieved significantly higher levels of grip strength (mean 41.8 kgf, 95% CI 40.7–42.8), with a 0.42 SD advantage over non-migrants (mean 38.6 kgf, 95% CI 38.3–39.0, $p < 0.001$) and a 0.30 SD advantage over internal migrants (mean 39.6 kgf, 95% CI 39.1–40.2, $p < 0.001$). Current international migrants were significantly more likely to have stage 1 or higher hypertension than non-migrants (13.0% versus 7.0%, $p = 0.014$), but they were no more likely than internal migrants.

## Multivariate analysis: Mental health

We estimated the association of migration status with mental health, measured by a 12-item shortened CES-D questionnaire. Fig 3 reports the kernel density estimates of aggregate depressive symptom scores by migrant status and mode of interview. These results suggest that the timing of Eid festival interviews may have biased the scores. Because these in-person interviews with international migrants were conducted at the one time in a 3-year period in which respondents were reunited with family and friends, overall CES-D scores were much lower than for any other group, including international phone survey respondents. We therefore excluded Eid festival interviews from analysis of mental health.

Table 5 reports the covariate-adjusted standardized depressive symptom scores by migrant status, using only the phone survey respondents to represent current international migrants. Using all 12 items, current international migrants had an adjusted mean score of 0.22 SD above the mean, with higher values indicating higher levels of depressive symptoms. Internal migrants showed significantly lower scores than international migrants (adjusted mean score −0.028 SD, $p < 0.001$), with non-migrants having a marginally significant advantage over international migrants (adjusted mean score 0.099 SD, $p = 0.090$). Looking only at the 4 items relating to positive emotions, there was little variation by migrant status, with current international migrants having an adjusted mean score of +0.098 SD and only internal migrants showing a marginally significant difference from international migrants (adjusted mean score

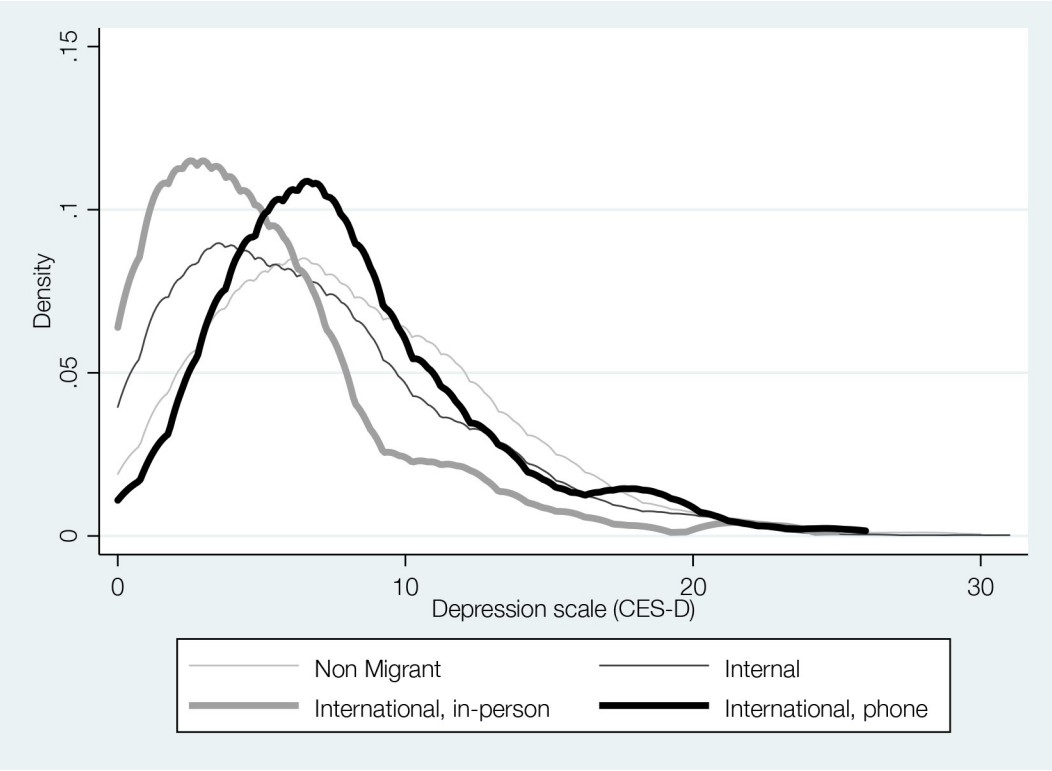

**Fig 3. Kernel density estimate of Center for Epidemiologic Studies Depression Scale (CES-D) scores by migrant status and mode of interview.** Kernel density estimates were constructed using the Stata kdensity function with Epanechnikov kernel and bandwidth of 1. For this analysis, migration status was based strictly on current residence, with returned international migrants included in the non-migrant or internal migrant groups, as appropriate. Sample sizes were as follows: non-migrant, 2,190; internal migrant, 1,315; current international migrant interviewed in person, 275; current international migrant interviewed by phone, 515.

**Table 5. Estimates of standardized depressive symptom scores by migration status—covariate-adjusted logistic regression models.**

| Status | Non-migrant | Internal migrant | Returned international migrant | Current international migrant[#] | n |
|---|---|---|---|---|---|
| All items (12) | 0.099[+] | −0.028[***] | 0.058 | 0.220 | 4,019 |
| | (0.043, 0.156) | (−0.111, 0.055) | (−0.121, 0.236) | (0.098, 0.342) | |
| Positive emotion items (4) | 0.111 | −0.027[+] | 0.105 | 0.098 | 4,019 |
| | (0.055, 0.168) | (−0.109, 0.055) | (−0.103, 0.313) | (−0.023, 0.219) | |
| Negative emotion items (8) | 0.056[**] | −0.020[***] | −0.003[*] | 0.254 | 4,019 |
| | (−0.003, 0.115) | (−0.099, 0.060) | (−0.236, 0.230) | (0.138, 0.370) | |

Data are adjusted mean score (SD) (95% CI). Marginal predictions from regressions controlling for age, education, religion, height, parental schooling, sibling composition, father's international migration, brother's international migration, and 1996–1997 household assets. Statistical test of difference from current international migrant

[***]$p < 0.001$

[**]$p < 0.01$

[*]$p < 0.05$

[+]$p < 0.10$.

[#]Excludes Eid festival survey respondents.

−0.028 SD, $p = 0.092$). Variations by migrant status were highly significant for items relating to negative emotions, however. International migrants had an adjusted mean score of +0.254 SD All other groups scored significantly lower than current international migrants, with internal migrants performing best (adjusted mean score −0.020 SD, $p < 0.001$). Non-migrants, internal migrants, and returned international migrants did not significantly differ from one another.

## Discussion

In this study, we observed that international migrants had comparable or lower injury and mortality risks compared to respondents remaining in Bangladesh, due in part to the high risks present in Bangladesh. International migrants also showed higher levels of self-rated health and physical strength, reflective of positive self-selection into migration. They had substantially higher body mass, higher risk of overweight/obesity, moderately higher risk of hypertension, and greater depressive symptoms.

Guest workers represent both a highly vulnerable population and critical yet understudied participants in the machinery of global development. Up to one-third of the world's international migrants are on temporary work visas. Several of the world's largest economies, including most of the destination countries included in this study, are highly dependent on guest workers. The mechanisms under which guest workers are recruited, employed, and often rejected are rightly debated on legal and ethical grounds [6]. Much of the narrative surrounding guest worker rights has been based on anecdotal and thinly sourced reports of widespread injury and fatality [37–39], alongside qualitative studies and non-representative studies pointing to other risks. By following a large sample of guest workers from within a larger prospective cohort study of migrants, internal migrants, and non-migrants, our study is the first that can adequately shed light on the health consequences of migration for guest workers working in the Persian Gulf and Asia.

Our results find little support for the most extreme concerns about injury and mortality. The most potent concern relates to the possibility that guest workers may face a substantial excess risk of mortality, whether in comparison to the those remaining in the sending country or to a global standard for working age populations. Mortality represents an important health outcome, and could also lead the results on non-mortality health outcomes in this paper to be

biased, as the most disadvantaged guest workers are missing from the data because they died. Our results appear to reject such concerns. Our attrition analysis yielded only 4 deaths among 945 individuals whose last known location was outside Bangladesh. Yet it remains possible, however unlikely, that respondents were injured abroad only to die in Bangladesh. We therefore took advantage of the prospective migration data from the MHDSS to conduct a survival analysis of the annual risk of mortality before and after individuals' first recorded episode abroad. We found that the annual hazard of mortality in the years following international migration was about 75% lower in years spent abroad than in years spent in Bangladesh, and comparable to average annual death probabilities for typical host country populations.

Relatedly, we found no evidence of excess injury risk among international migrants, and in fact found a modest advantage relative to non-migrants. But this does not mean that injury risks were low among international migrants, but rather that the risks of injury are extraordinarily high in Matlab and throughout Bangladesh [44]. We found no significant differences in terms of workplace injuries. Given the wide economic and infrastructural disparities between guest worker destinations and major Bangladeshi cities like Dhaka, we might expect substantially fewer injuries among international migrants, and so the lack of a migrant advantage in this case may still raise concern.

Yet a much clearer picture of the health burdens affecting guest workers emerges when we consider chronic disease and associated risk factors. International migrants show numerous health advantages that largely reflect the substantial force of self-selection into the physically taxing process of migration. International migrants not only come from socioeconomically advantaged households in terms of wealth and schooling, they also have substantial advantages in terms of stature and strength. Based on reports for the full sample, international workers on average are about 2 cm taller than non-migrants, have grip strength 0.4 SD higher, and are half as likely to report fair/poor self-rated health. They are also less likely to smoke, though this advantage was diminished by control variables. Each of these tendencies has been documented in studies of migrant selection using paired or matched samples from sending and host countries [8,11], but such studies always carry the possibility of measurement or sampling bias across distinct samples. Our analysis based on a population follow-up study confirms evidence of positive self-selection from earlier studies.

In spite of these obvious health advantages, international migrants also bear markers of health disadvantage in terms of dietary and metabolic risk, and to a lesser extent mental health. Guest workers have a mean BMI that is 2.3 kg/m$^2$ higher than that of their non-migrant peers (a 0.7 SD difference) and 1.3 kg/m$^2$ higher than that of internal migrants. As a result, guest workers are twice as likely as non-migrants or internal migrants to be obese according to the Asian standard cutoff of BMI $\geq 27.5$ kg/m$^2$. The 7% adjusted obesity prevalence among international migrants is still relatively low in comparison to host populations or to international migrants in other societies. Yet we observe a far more substantial attributable burden of overweight/obesity among international migrants, who have a 52% adjusted risk of being overweight or obese compared to just 23% among non-migrants and 38% among internal migrants. Given the secular trend towards rising metabolic risk in Bangladesh [45] and the high susceptibility to diabetes and cardiovascular disease among South Asian populations [46], these differences may point to serious health risk. This concern is further reinforced by the fact that international migrants were nearly twice as likely to experience stage 1 or higher hypertension as non-migrants, though we note that internal migrants had equally high hypertension burdens. We also observed significantly higher depressive symptom scores among international migrants, particularly when focusing on those CES-D items that address negative emotions. While it is salutary that burdens of hypertension and mental illness appear to diminish substantially when migrants return home permanently or for festivals, they nonetheless raise

cause for concern that should be addressed in long-term follow-up studies of health and mortality.

This initial analysis carries a number of limitations, some of which can be addressed in future work. First, this study is merely descriptive. Future work based on a follow-up survey currently in the field will tie variations in health to variations in the conditions of entry into host countries (social networks, debt, living conditions), to occupational risk exposures (measured by frequency and protection), and experiences of abuse. Second, while our data provide valuable controls for baseline characteristics dating back decades, the long gap between baseline and follow-up surveys and the fact that many respondents were very young at baseline does not allow us to directly measure a change in any single health indicator associated with migration. Third, the use of a combination of in-person interviews during Eid festivals and phone surveys for international migrants created important gaps in the availability of objective health measures for phone respondents, in the need to use a shortened 12-item CES-D module, and in the validity of mental health data for festival respondents. This also raises concerns about the validity and comparability of measures drawn from distinct survey modes, though a methodological companion paper demonstrates the high level of comparability of income, height, and weight reports [47]. Finally, the study would benefit from more salient and time-sensitive measures of injury, abuse, and mental distress. Many of the most salient risks affecting guest workers can occur on a very rapid temporal scale, whether we are speaking of daily variations in heat stress, episodes of abuse, or visa issues. In the future we hope to link real-time measures of risk exposure with ecological momentary analysis of mental distress using a mobile data collection platform.

Finally, we note that temporary guest worker migration is a diverse process, containing multiple migrant streams that are stratified by industry, recruitment pathway, and exposure to risk. Matlab has a very well established migration history and well-developed recruitment mechanisms, and so results might look different in source populations with lower stocks of migration-specific social capital.

In spite of these limitations, this study raises critical policy concerns with respect to the health of guest workers and points the way to a conceptual model of guest worker health, and immigrant health more broadly. Guest worker migration continues to be popular because, in spite of the risks, we find that migrants earn roughly 4 times as much annually as non-migrants, and 3 times as much as internal migrants. In the process, their health may be compromised by the physical burdens of their work and by the psychological burdens of separation from family and home society. Migrant health may also be influenced by the complex mix of assimilation into local dietary and health-seeking behavior patterns but exclusion from key societal and health system functions. In spite of these burdens, our findings do not support the notion that guest worker migration should be eliminated or curtailed simply on the basis of health risks. Our qualitative interviews with migrants revealed that migrants are drawn to migration, in spite of the well-understood risks, by the opportunity to secure their own livelihoods, improve their family security, and also explore a new society and culture.

Both the countries of arrival and departure can respond to these findings by introducing measures to reduce the intensity of work-related stressors, societal adjustment issues, and family separation. These can include better working and living conditions and shorter working hours, as well as more flexible family leave arrangements. The Global Compact for Safe, Orderly and Regular Migration offers a template for pursuing these goals on a multilateral basis [48]. Implementation of these goals will require further binational monitoring of migrant health and well-being along with interventions and policy analysis aimed at identifying best practices for protecting migrant health.

## Conclusion

This descriptive analysis of the health and well-being of current and returned international guest workers from a rural area of Bangladesh provides rare evidence on a population that is both understudied and at risk. The comparison of migrants to non-migrant and internal migrant comparators, combined with longitudinal data going back many years, allows us to approach an answer to the question of whether guest worker migration is bad for health, and in what ways. Contrary to the popular narrative of a population saddled with acute health risks relating to injury and mortality, our results point to a more subtle set of emerging health risks that nonetheless raise concern. International migrants show significantly better health on measures of long-term health such as grip strength, reflecting self-selection in the highly competitive process of migration. Yet in spite of these advantages, migrants have significantly higher risks of overweight, obesity, hypertension, and depression than both non-migrant and internal migrant comparators. While returned international migrants in some cases look more similar to non-migrants, international migrants nonetheless accumulate years of exposure to chronic disease and mental health risk. The new Global Compact for Safe, Orderly and Regular Migration offers a potential mechanism for addressing these burdens through bilateral cooperation.

## Supporting information

**S1 STROBE Checklist. STROBE statement.**
(DOC)

**S1 Table. Full model specifications for all dependent variables.**
(DOCX)

**S1 Text. Proposal submitted to Research & Empirical Analysis of Labor Migration (REALM).**
(DOCX)

## Acknowledgments

The authors gratefully acknowledge the contributions of Mitra and Associates fieldwork staff, especially Shahidul Islam, Zahidul Islam, Shamim Hasan Habib, and Syed Abdullah Al Ahsan. We thank icddr,b staff who supported fieldwork, especially Nazrul Islam, Monoranjan Das, A. H. M. Golam Mustafa, and Taslim Ali. Graduate students who supported this work include Sveta Milusheva, Gisella Kagy, Bryan Phillips, and Zhenxiang "Zeke" Chen. The work would not have been possible without the administrative support of Steve Graham, Preethi Thomas, Melba Tolbert, and Gloria Greengard. The authors received helpful feedback from Fernando Riosmena, Roger Waldinger, and members of the Research & Empirical Analysis of Labor Migration (REALM) workshop. Earlier versions of this paper were presented at the California Center for Population Research, UCLA Center for the Study of International Migration, Carolina Population Center, Population Association of America, and the International Forum on Migration Statistics.

## Author Contributions

**Conceptualization:** Randall Kuhn, Tania Barham, Abdur Razzaque.

**Data curation:** Randall Kuhn, Tania Barham, Patrick Turner.

**Formal analysis:** Randall Kuhn.

**Funding acquisition:** Randall Kuhn, Tania Barham.

**Investigation:** Randall Kuhn, Tania Barham, Abdur Razzaque, Patrick Turner.

**Methodology:** Randall Kuhn, Tania Barham.

**Resources:** Abdur Razzaque.

**Validation:** Abdur Razzaque.

**Writing – original draft:** Randall Kuhn.

**Writing – review & editing:** Randall Kuhn, Tania Barham, Abdur Razzaque, Patrick Turner.

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
