## [Decision Letter · Decision Letter 0]

18 Dec 2019

Dear Dr. Kuhn,

Thank you very much for submitting your manuscript "Is being a guest worker bad for your health? Lessons from a large-scale binational survey of Bangladeshi migrants and non-migrants" (PMEDICINE-D-19-03799) for consideration at PLOS Medicine for our upcoming special issue on refugee and migrant health. 

Your paper was discussed among the editorial team and sent to independent reviewers, including a statistical reviewer. The reviews are appended at the bottom of this email and any accompanying reviewer attachments can be seen via the link below:

[LINK]

In light of these reviews, we will not be able to accept the manuscript for publication in the journal in its current form, but we would like to invite you to submit a revised version that fully addresses the reviewers' and editors' comments. You will appreciate that we cannot make a decision about publication until we have seen the revised manuscript and your response, and we expect to seek re-review by one or more of the reviewers. 

We hope to receive your revised manuscript by January 6th. Please email us (plosmedicine@plos.org) if you have any questions or concerns.

Please let me know if you have any questions. Otherwise, we look forward to receiving your revised manuscript soon. 

Sincerely,

Richard Turner, PhD

rturner@plos.org

Please finalize the arrangements for data deposition.

Regarding data access, you state "No: some restrictions will apply", and we ask you to detail these restrictions. 

We ask you to restructure your title to accord with journal style (non declarative, with a study descriptor). We suggest: "Health and wellbeing of international migrants and non-migrants in Bangladesh: a cross-sectional study". 

Please combine the "methods" and "findings" subsections of your abstract; and add a new final sentence to the new combined subsection summarizing the study's main limitations. 

In the abstract, please detail study participants along with summary demographic details. Please also provide quantitative estimates for the main study outcomes, along with 95% CI and p values where appropriate. 

Please begin the "conclusions" subsection of your abstract with "In this study, we observed that ..." or similar, and condense the subsection.

After the abstract, we will need to ask you to add a new and accessible "author summary" section in non-identical prose. You may find it helpful to consult one or two recent research papers in PLOS Medicine to get a sense of the preferred style. 

We ask you to trim the introduction section of your main text. Around 500 words is usually sufficient; additional material on previous literature can perhaps be relocated to the discussion section. 

Please avoid claims of "the first" or "largest", and where necessary add "to our knowledge" or similar. 

Early in the methods section of your main text, please state whether the study had a protocol or prospective analysis plan, and if so attach the document(s) as a supplementary file (referred to in the text). Please highlight analyses that were not prespecified. 

Please add a brief statement to the methods section on study ethics approval, whether required or not. 

Please restructure the early part of the discussion section of your main text so that the first paragraph provides a summary of the paper's findings. 

Please format reference call-outs as follows: "... in 2018 [1,4].".

We ask that you revisit the reference list to ensure that all citations meet journal style. Reference 41 may need additional access information to be included. 

Please add a completed checklist for the most appropriate reporting guideline, which might be STROBE or RECORD, as a supplementary document (referred to in your methods section). In the checklist, please refer to individual items by section (e.g., "Methods") and paragraph number rather than by page or line numbers, as the latter generally change upon publication. 

Comments from the reviewers:

*** Reviewer #1: 

I confine my remarks to statistical aspects of this paper. The general approach is fine, but I have a few issues to resolve before I can recommend publication.

NOTE: Adding page and line numbers would have made it easier to reference my comments

General comment: Why not use some sort of matching (e.g. by propensity score)? I'm not saying it's required, but was it considered? If so, why was it rejected?

Abstract:

It's logistic regression, not logistics

There should only be one reference group.

Introduction:

"larger representative sample ..." representative of what?

Methods

The formula can probably be deleted. As written, it is incorrect in that some fo the betas are vectors, while the formula implies they are scalars. You would need one beta for each variable, not for groups of variables

Independent variables (e.g. BMI) should not be categorized. In *Regression Modeling Strategies* Frank Harrell lists 11 problems that this gives rise to and summarizes "nothing could be more disastrous". I wrote a blog post illustrating some of these, graphically: https://medium.com/@peterflom/what-happens-when-we-categorize-an-independent-variable-in-regression-77d4c5862b6c

When listing all the IVs, they need to be given operational definitions. That is how were they measured? e.g. what were the choices for "religion"?

Why were fair and poor health lumped together? This is probably a mistake.

Table 5: Give mean BMI and grip strength for each group. Don't categorize. 

Peter Flom

*** Reviewer #2: 

This study represents important advances in research of migrant health. There are very few studies that are able to compare migrants, non-migrants and returned migrants from the same community; this study does so using very high quality longitudinal data from a long-standing Demographic Surveillance System. The focus of the study is on migration from South Asia to the Middle East - one of the very established routes for labour migration, but also a very dangerous one for migrants, entailing brutal work conditions, lack of access to services, servitude to an employer, and high risks of death and injury.

The methods and data are appropriate. I have a few questions relating to models and interpretation.

1. The paper highlights higher risks of depression among migrants, but these seem to be non-significant in Table 6.

2. With respect to the models, it would be useful to first estimate an interaction model, in which all respondents are pooled and migration status is interacted, for example with sadness items. This would allow the authors to directly estimate the effect of migrations status. It would also indicate where stratified analyses are needed.

3. When we think about poor health of guest workers, we generally think about injuries and harm directly resulting from the labour migration. This paper does not find evidence of such problems, but instead finds evidence of overweight and hypertension (though this is also not significant). These are different types of health risks, and are not directly hazards of the work migrants are doing, but likely rather a bi-product of integration into the host society. It would be informative if the article also indicated the levels of overweight in the host country - do migrants just look like the population into which they are integrating? If so, this should be discussed as a component of integration foremost, and less so as a hazard of migration.

4. The interpretation in the abstract goes further than that in the paper in describing the depletion of migrants, and seems to go beyond the findings of the paper.

5. It seems there is something interesting going on with internal migrants. Either in this paper or another, it will be an interesting to examine.

*** Reviewer #3: 

This study benefits from well described migrant vs non-migrant groups with adequate follow-up over time. That is a unique and major strength of the study. Below some feedback to constructively contribute to improve this work.

MAJOR COMMENTS

1. Methods: "Along with predicted values/probabilities and confidence intervals, we report statistical tests of difference in which current international migrants are the reference category, thereby comparing international migrants to all three counterfactual comparisons groups of non-migrants, internal migrants, and returned migrants." Returned migrants are not described in Table 1. The reason I mention this is because returned migrants can be internal or international returning migrants, so clarification is needed. Later on, it reads "Among respondents in Bangladesh, returned international migrants were identified based on a self-report of whether they had lived abroad in the past five years..." Best to clarify throughout the text.

2. Table 1 would be better to be transformed into a Flowchart, to comply with STROBE recommendations, and in that flowchart add how many people were evaluated by what method (in-person or telephone) and how much of the actual data was measured vs self-reported.

3. Depression. In methods the authors say that "Because there is no validated diagnostic cutoff for depression using this abridged scale, we estimated models for standardized depression scores." What were those scores? How do they compare or related to the existing literature? Are they comparable to other definitions of depression and the cut-offs used? Possibly, given these differences, the authors are better off not labelling this as depression (which has a gold standard definition) but as depressive symptoms. All of this should be clear in methods and also described as a limitation.

4. Hypertension. Why do the authors used 130/80 mm Hg as the cut offs? Explain in the manuscript. Is that cut off applicable to Bangladesh population?

5. On mortality. When the authors say, in the Discussion, that "This does not mean that injury risks were low among international migrants, but rather that the risks of injury are extraordinarily high in Matlab and throughout Bangladesh (41)", this reviewer wonders if a similar reasoning could be applied to the mortality indicator. Also, on the mortality, is is something that would require longer follow-up periods to observe differences? What if the accumulated risk factors —higher obesity, hypertension and depression scores among international migrants— would turn into higher mortality later on? Some reflection on this would be ideal.

MINOR COMMENTS

6. Introduction and discussion. It could benefit from more recent bibliography. See at the end some suggested references for the authors to consider.

7. Some clarification of the type of location of internal migration and how much similar were they to Matlab? This is important to understand, a bit, the context of the internal migrants.

8. On the reference group. See comment #1 and contrast it with this sentence from Findings: "Table 3 compares the baseline characteristics of migration status groups. On most characteristics, international and internal migrants were comparable, making internal migrants a relevant reference group." Avoid confusions.

9. Typo: mich. "International migrants were mich more likely to have a father who had lived abroad."

10. Consider moving Fig 1 (bivariate analysis) to supplementary material and transform some of the tables in graphs for a better communication of the estimates presented.

SUGGESTED REFERENCES

1: Dominguez K, Penman-Aguilar A, Chang MH, Moonesinghe R, Castellanos T, Rodriguez-Lainz A, Schieber R; Centers for Disease Control and Prevention (CDC). 

Vital signs: leading causes of death, prevalence of diseases and risk factors, and use of health services among Hispanics in the United States - 2009-2013. 

MMWR Morb Mortal Wkly Rep. 2015 May 8;64(17):469-78. Erratum in: MMWR Morb Mortal Wkly Rep. 2015 Oct 16;64(40):1153. PubMed PMID: 25950254; PubMed Central PMCID: PMC4584552.

2: Abubakar I, Aldridge RW, Devakumar D, Orcutt M, Burns R, Barreto ML, Dhavan P, Fouad FM, Groce N, Guo Y, Hargreaves S, Knipper M, Miranda JJ, Madise N, Kumar B, Mosca D, McGovern T, Rubenstein L, Sammonds P, Sawyer SM, Sheikh K, Tollman S, Spiegel P, Zimmerman C; UCL-Lancet Commission on Migration and Health. 

The UCL-Lancet Commission on Migration and Health: the health of a world on the move.

Lancet. 2018 Dec 15;392(10164):2606-2654. doi: 10.1016/S0140-6736(18)32114-7. Epub 2018 Dec 5. Review. PubMed PMID: 30528486.

3: Aldridge RW, Nellums LB, Bartlett S, Barr AL, Patel P, Burns R, Hargreaves S, Miranda JJ, Tollman S, Friedland JS, Abubakar I. 

Global patterns of mortality in international migrants: a systematic review and meta-analysis. 

Lancet. 2018 Dec 15;392(10164):2553-2566. doi: 10.1016/S0140-6736(18)32781-8. Epub 2018 Dec 5. PubMed PMID: 30528484; PubMed Central PMCID: PMC6294735.

4: Hargreaves S, Rustage K, Nellums LB, McAlpine A, Pocock N, Devakumar D, Aldridge RW, Abubakar I, Kristensen KL, Himmels JW, Friedland JS, Zimmerman C.

Occupational health outcomes among international migrant workers: a systematic review and meta-analysis. 

Lancet Glob Health. 2019 Jul;7(7):e872-e882. doi: 10.1016/S2214-109X(19)30204-9. Epub 2019 May 20. PubMed PMID: 31122905; PubMed Central PMCID: PMC6565984.

5: Juárez SP, Honkaniemi H, Dunlavy AC, Aldridge RW, Barreto ML, Katikireddi SV, Rostila M. 

Effects of non-health-targeted policies on migrant health: a systematic review and meta-analysis. 

Lancet Glob Health. 2019 Apr;7(4):e420-e435. doi: 10.1016/S2214-109X(18)30560-6. Epub 2019 Mar 6. PubMed PMID: 30852188; PubMed Central PMCID: PMC6418177.

6: Lu Y, Qin L. 

Healthy migrant and salmon bias hypotheses: a study of health and internal migration in China. 

Soc Sci Med. 2014 Feb;102:41-8. doi:10.1016/j.socscimed.2013.11.040. Epub 2013 Nov 28. PubMed PMID: 24565140.

7: Puschmann P, Donrovich R, Matthijs K. 

Salmon Bias or Red Herring? : Comparing Adult Mortality Risks (Ages 30-90) between Natives and Internal Migrants: Stayers, Returnees and Movers in Rotterdam, the Netherlands, 1850-1940. 

Hum Nat. 2017 Dec;28(4):481-499. doi: 10.1007/s12110-017-9303-1. PubMed PMID: 29043501; PubMed Central PMCID: PMC5662680.

8: Dodd W, Humphries S, Patel K, Majowicz S, Little M, Dewey C. 

Determinants of internal migrant health and the healthy migrant effect in South India: a mixed methods study. 

BMC Int Health Hum Rights. 2017 Sep 12;17(1):23. doi:10.1186/s12914-017-0132-4. PubMed PMID: 28899374; PubMed Central PMCID: PMC5596496.

9: Hayes L, White M, McNally RJQ, Unwin N, Tran A, Bhopal R. 

Do cardiometabolic, behavioural and socioeconomic factors explain the 'healthy migrant effect' in the UK? Linked mortality follow-up of South Asians compared with white Europeans in the Newcastle Heart Project. 

J Epidemiol Community Health. 2017 Jul 25. pii:jech-2017-209348. doi: 10.1136/jech-2017-209348. [Epub ahead of print] PubMed PMID: 28743730.

***

[LINK]

---

## [Decision Letter · Decision Letter 1]

6 Feb 2020

Dear Dr. Kuhn,

Thank you very much for re-submitting your manuscript "Health and wellbeing of international migrants and non-migrants in Bangladesh: a cross-sectional followup study" (PMEDICINE-D-19-03799R1) for consideration at PLOS Medicine for our upcoming special issue on refugee and migrant health.

I have discussed the paper with editorial colleagues and the guest editors for the special issue, and it was also seen again by two reviewers. I am pleased to tell you that, provided the remaining editorial and production issues are dealt with, we expect to be able to accept the paper for publication in the journal.

[LINK]

We hope to receive your revised manuscript within 5 days, with apologies for the tight timeline. Please email us (plosmedicine@plos.org) if you have any questions or concerns.

Please let me know if you have any questions. Otherwise, we look forward to receiving the revised manuscript shortly. 

Sincerely,

Richard Turner, PhD

rturner@plos.org

Requests from Editors:

We ask you to finalize the arrangements for data deposition. 

Please add "male", or similar, to the title.

At line 40, please make that "... a single out-migration area of origin.".

At line 46, we suggest substituting "ill health" for "weathering". 

Rather than "current international migrants" which to our understanding does not reflect the "current" situation (in the abstract and elsewhere), we suggest rephrasing to "at follow-up, international migrants ..." or similar. 

In the abstract and elsewhere, please quote exact p values or p< 0.001.

We ask you to trim the "author summary" section. Generally, we expect the three subsections to contain 2-3 points, each consisting of 1-2 short sentences. Background information and information on earnings, for example, can be substantially reduced. 

Please refer to the attached STROBE checklist in the methods section of your main text. 

At line 475, please revisit the phrase "temporary guest worker is a diverse process" to improve readability. 

At line 493, we suggest rewording to "Both countries of departure and arrival ..." or similar. 

At line 507, we suggest removing "your" (from "your health") as you are referring to migrant health.

Please adopt the style "follow-up" throughout the text. 

Please trim the journal name for reference 42 to "Lancet". 

Comments from Reviewers:

*** Reviewer #1: 

The authors have addressed my concerns and I now recommend publiication

Peter Flom

*** Reviewer #2: 

The authors have been responsive to reviewers and have engaged very actively with the comments received - this has further improved an excellent paper.

My one substantial comment:

Are there differences in health-relevant characteristics among who goes to the various destinations? As you summarize the findings, are there differences in experiences of guest workers and their health outcomes depending on where they go that should be mentioned?

Other thoughts for your consideration:

When you say "also examined observed health measures", be clearer that you actually measured hypertension. Would be important to note when you are talking about self-reported vs. directly-measured indicators. For example, was BMI measured for some people and self-reported for others? If so, there could be systematic differences to consider.

In terms of BMI, you focus on overweight/obese, but was there also underweight, eg. among non-migrants?

It is really nice that you are able to compare some measures for in-person interviews vs. phone interviews, and interesting to see how these differ. Thoughts on why?

The figures are a bit blurry.

***

[LINK]

---

## [Editor Report · Decision Letter 2]

27 Feb 2020

Dear Prof. Kuhn, 

On behalf of my colleagues and the academic editor, Dr. Paul Spiegel, I am delighted to inform you that your manuscript entitled "Health and wellbeing of male international migrants and non-migrants in Bangladesh: a cross-sectional followup study" (PMEDICINE-D-19-03799R2) has been accepted for publication in PLOS Medicine. 

PRODUCTION PROCESS

PRESS

PROFILE INFORMATION

Thank you again for submitting the manuscript to PLOS Medicine. We look forward to publishing it. 

Best wishes, 

Richard Turner, PhD

Senior Editor 

PLOS Medicine

plosmedicine.org